# Management of Coronary Artery Disease in CADASIL Patients: Review of Current Literature

**DOI:** 10.3390/medicina59030586

**Published:** 2023-03-16

**Authors:** Maria Servito, Isha Gill, Joshua Durbin, Nader Ghasemlou, Aron-Frederik Popov, Christopher D. Stephen, Mohammad El-Diasty

**Affiliations:** 1Department of Cardiac Surgery, University of Manitoba, Winnipeg, MB R2H 2A6, Canada; 2Department of Biomedical and Molecular Sciences, Queen’s University, Kingston, ON K7L 2V7, Canada; 3Division of Cardiology, Department of Medicine, Queen’s University, Kingston, ON K7L 2V7, Canada; 4Department of Anaesthesiology, Queen’s University, Kingston, ON K7L 2V7, Canada; 5Department of Cardiothoracic Surgery, Helios Clinic, 53721 Siegburg, Germany; 6Department of Neurology, Massachusetts General Hospital, Harvard Medical School, Boston, MA 02114, USA; 7Division of Cardiac Surgery, Department of Surgery, Queen’s University, Kingston, ON K7L 2V7, Canada

**Keywords:** coronary artery disease, CADASIL, stroke, cerebral microbleeds

## Abstract

Cerebral autosomal dominant arteriopathy with subcortical infarcts and leukoencephalopathy (CADASIL) is the most common heritable form of vascular dementia in adults. It is well-established that CADASIL results in neurocognitive dysfunction and mood disturbance. There is also cumulative evidence that CADASIL patients are more susceptible to ischemic heart disease. The aim of this study is to review the current literature regarding the incidence of coronary artery disease in CADASIL patients with a focus on the various management options and the clinical challenges associated with each of these treatment strategies. We conducted a literature search using Cochrane, MEDLINE, and EMBASE for papers that reported the occurrence of coronary artery disease in patients with CADASIL. We supplemented the search with a manual search in Google Scholar. Only case reports, case series, and original articles were included. The search resulted in six reports indicating the association between coronary artery disease and CADASIL and its management. Evidence suggests that extracranial manifestations of CADASIL may include coronary artery disease, presenting as a more extensive burden of disease in younger patients. Surgical and percutaneous revascularization strategies are feasible, but the incidence of peri-procedural stroke remains significant and should be weighed against the potential benefit derived from either of these strategies. A multidisciplinary approach to therapy, with perspectives from neurologists, cardiologists, and cardiac surgeons, is needed to provide the appropriate treatment to the CADASIL patient with severe coronary artery disease. Future studies should be directed toward the development of targeted therapies that may help with the early detection and prevention of disease progress in these patients.

## 1. Introduction

Cerebral autosomal dominant arteriopathy with subcortical infarcts and leukoencephalopathy (CADASIL) is a heritable disease that was first described in 1955 and, since then, has been shown to result in neurological dysfunction and disability in middle-aged individuals [1,2]. CADASIL commonly manifests as lacunar infarcts as detected by magnetic resonance imaging (MRI) of the brain [1], but cerebral microhemorrhages can also be detected in up to 25–70% of CADASIL patients, depending on the age of presentation [3]. While commonly reported to primarily manifest as neurocognitive dysfunction and decline, patients with CADASIL may also experience seizures, encephalopathy, and mood disorders, owing to the demyelination and axonal damage that occur due to the genetic mutations [1]. Non-neurological manifestations of CADASIL have also been reported, including increased risk of peripheral vascular disease, retinal artery disease, and renal artery involvement [2]. Also, there is an increasing number of reports of CADASIL patients who present with ischemic coronary artery disease (CAD) [2]. The pathophysiology of CAD in CADASIL patients is complex, and while many patients present with obstructive atherosclerotic coronary lesions, some patients may present with non-atherosclerotic disease mainly due to coronary microvascular dysfunction [4,5]. The clinical pattern of CAD in CADASIL patients is usually characterized by the early onset of symptoms and a tendency to recurrence of coronary events [6,7]. The aim of this review is to discuss the current evidence regarding the incidence of coronary artery disease in CADASIL patients, with particular emphasis on the clinical presentation and early diagnosis of this condition. In addition, we present the different management options and the challenges associated with different revascularization strategies in this group of patients.

## 2. Search Strategy

We conducted an extensive English-language literature search using Cochrane, MEDLINE, and EMBASE for all papers that reported the occurrence of CAD in patients with CADASIL. We included all articles irrespective of the article type. The search was subsequently supplemented with a manual search in Google Scholar. All articles were reviewed, and only relevant studies were included (Table 1).

## 3. Epidemiology

CADASIL is considered to be the most prevalent type of hereditary vascular dementia in the adult population [7]. According to the literature, the prevalence of CADASIL is estimated to be 4.15 per 100,000 individuals [13,14,15]. However, this is likely to be underestimated as many patients remain underdiagnosed, and de novo cases may occur [1]. In addition to dementia, other presenting symptoms may include apathy and mood disorders [7]. The clinical presentation of CADASIL may also vary depending on the sex of the affected individual; while migraines with aura are 50% more prevalent in women, stroke is 75% more prevalent in men [16]. Although the neurological manifestations of CADASIL are well described, the incidence of CAD in patients with CADASIL has not been fully studied [8,17]. This is mainly due to the infrequent incidence of these conditions and the lack of large registries. Published reports mostly consist of case reports and small case series (Table 1) [6,8,9,10,11,12]. However, these reports suggest that CADASIL may play an important role in the pathogenesis and development of a severe and more aggressive and recurrent form of CAD [3,4,5,7,18,19].

## 4. Pathophysiology

CADASIL is an autosomal dominant disorder caused by a mutation in the *NOTCH3* gene [1]. In most cases, an affected individual has an affected parent, but in rare instances, de novo mutations can also occur [7]. *NOTCH3* is highly expressed in vascular smooth muscle cells, and its function has important implications for the maintenance of vascular contractility [1]. Patients with CADASIL have a gain-of-function mutation in *NOTCH3* that results in the deposition of microscopic aggregates around the smooth muscle cells with the characteristic appearance of granular osmiophilic material (GOM) on electron microscopic studies [1]. Nevertheless, the deposition of these aggregates results in vascular fibrosis and smooth muscle degeneration [20], impaired vasoreactivity of small arteries [20], increased myogenic tone [13], increased vascular thickening [13], and eventually luminal stenosis [21].

Therefore, these mutations result in vasculopathy affecting small and medium-sized cerebral arteries and arterioles [4,13]. Consequently, impaired autoregulation of cerebral blood flow leads to a spectrum of clinical disorders that include the premature onset of small vessel ischemic disease, cognitive decline with vascular dementia, psychiatric disorders, and migraine with aura [1,4,13,21].

Extracerebral manifestations of the *NOTCH3* mutation have also been described. For instance, this mutation can result in diffuse subendocardial hypoperfusion manifesting as angina in the absence of atherosclerotic coronary obstruction due to the deposition of the GOM aggregates within the coronary vascular wall [11]. Pathological studies in CADASIL patients show subendocardial fibrosis and thickening without evidence of significant atherosclerotic or thrombotic changes in the coronary vasculature [5]. The occurrence of significant perivascular fibrosis, elastosis, and cellular degeneration has also been reported [5]. Indeed, published reports suggest that coronary microvascular dysfunction in these patients is primarily attributed to fibrous vessel wall thickening, impaired vasoreactivity, and increased myogenic tone [13,20]. Consequently, compromise of myocardial perfusion occurs owing to the inability to increase coronary flow rates in response to increased oxygen demands [5,20,22]. This mechanism may explain why some CADASIL patients may present with typical coronary ischemic changes in absence of significant coronary obstructive lesions on diagnostic imaging modalities [4].

In addition, it was suggested that CADASIL patients might be predisposed to a premature form of CAD that is characterized by its early onset and diffuse nature of coronary lesions even in the absence of typical coronary artery risk factors such as arterial hypertension, hypercholesterolemia, and positive family history for CAD. Therefore, current cardiovascular risk prediction models, such as the Atherosclerotic Cardiovascular Disease (ASCVD) risk estimator, may not apply to this group of patients [7].

## 5. Cardiac Manifestations of CADASIL

The association between CAD and CADASIL has been described in multiple reports (Table 1), but the evidence remains limited, and thus, the true prevalence of CAD among patients with CADASIL remains unknown. Nevertheless, there is a consistent pattern among these studies that suggests the premature onset and aggressive nature of CAD in CADASIL patients.

A common finding in CADASIL patients with CAD is the non-atherosclerotic nature of the disease. The concept of Angina in the absence of obstructive coronary disease (ANOCA) was highlighted in the case report by Langer and colleagues, wherein a coronary CT angiogram performed on a patient with CADASIL revealed diffuse vessel wall irregularities, vascular thickening, and mixed non-calcified/calcified coronary lesions with intermediate stenosis. This was consistent with the evidence of subendocardial hypoperfusion from a noninvasive study in the affected coronary territories [11]. A case series of 17 patients with CADASIL further confirmed these findings, whereby positron emission tomography showed impaired coronary blood flow and coronary flow reserve with areas of diffuse hypoperfusion [22].

As a result of the coronary microvascular dysfunction in CADASIL patients, CAD is more aggressive and accelerated, wherein patients present not only at a younger age but also with a more extensive burden of disease. For instance, Oberstein and colleagues found that in the five patients who had a history of acute MI, the mean age at the time of the first event was 39.6 years old [5]. Noteworthy, their coronary angiogram showed an absence of flow-limiting stenotic lesions [5]. Instead, diffuse narrowing of coronary vasculature seems to be the typical feature [6,12,23]. In their case report, Ruben and colleagues described de novo CADASIL in a 45-year-old patient who had two episodes of unstable angina, both requiring drug-eluting coronary stents, a history of multiple transient ischemic attacks, and chronic headaches [6]. The patient neither had a family history of cardiac disease nor did she have significant cardiovascular risk factors [6]. When a coronary angiogram was performed, it indeed showed no focal stenosis but only a diffuse and irregular narrowing of the involved coronary arteries, which were the left anterior descending and right coronary arteries [6].

Also, spontaneous coronary artery dissection (SCAD) has been reported in the context of CADASIL [9,24]. The exact role of CADASIL in the pathogenesis of SCAD has not been established, but it may be due to vascular wall weakness secondary to vascular fibrosis and impaired autoregulation, as was suggested by Tsanaxidis and colleagues in their case study [9]. However, more robust evidence is clearly warranted to establish the role of CADASIL in the pathogenesis of SCAD.

It is important to mention that most of the current evidence was derived from case reports and small case series. Nevertheless, there is evidence to suggest that CADASIL may accelerate and exacerbate the nature of CAD in the absence of cardiovascular risk factors.

## 6. Diagnosis

The classic symptoms of CADASIL include recurrent strokes, cognitive impairment, migraines with or without aura, and psychiatric disturbances. When CADASIL is suspected based on the constellation of the abovementioned symptoms, the diagnosis is confirmed with either molecular genetic testing or skin biopsy [7]. There are two methods of molecular genetic testing to establish the diagnosis: single-gene testing or comprehensive gene testing, whereby the latter is preferred when the clinical phenotype is indistinguishable from other heritable disorders, such as vascular dementia [7]. When the findings of molecular gene testing are indeterminate, a skin biopsy may be indicated. Immunohistochemistry of the biopsied tissue is performed to detect the GOM within the blood vasculature, which is pathognomonic and confirmatory of CADASIL [7].

Brain MRI may also show symmetric and confluent white matter disease, particularly involving the anterior temporal lobes, which is an uncommon location for small vessel disease [1]. Other findings on imaging also include multiple subcortical ischemic infarcts related to micro-vessel angiopathy, dilated perivascular spaces, and cerebral atrophy [18]. Although the overall incidence of spontaneous intracranial hemorrhage (ICH) in CADASIL patients is low, cerebral microbleeds are relatively common [9,23,25,26,27,28].

In general, the diagnosis of CAD in CADASIL patients follows the same guidelines and diagnostic modalities as in non-CADASIL patients. To establish the diagnosis of coronary artery disease, a variety of invasive and non-invasive anatomic and functional assessment tests may be indicated to determine the severity and extent of myocardial ischemia. In patients presenting with symptoms suggestive of myocardial ischemia, in which an invasive diagnostic strategy is pursued, epicardial coronary artery stenotic lesions can be visualized utilizing selective coronary angiography [29]. Using well-established fluoroscopic techniques, the degree of coronary arterial blood flow limitation can be estimated using visual criteria. In the presence of a borderline epicardial coronary artery stenosis in which the degree of severity of blood flow limitation is not evident or equivocal, the use of invasive physiologic testing under hyperemic and non-hyperemic conditions may be indicated. Techniques such as fractional flow reserve (FFR), as assessed in the FAME and FAME 2 trials, or instantaneous free wave ratio (iFR), as assessed in the iFR-SWEDEHEART trial, have proven helpful in guiding revascularization strategies [30,31,32]. Comprehensive guidance for the general assessment and workup of coronary artery disease and myocardial ischemia is provided by major societal guidelines [33] and is out of scope for this review.

It is also very important to distinguish between patients with coronary artery disease amenable to revascularization, such as those with focal epicardial stenosis resulting in discrete areas of flow limitation, as defined by angiographic appearance and iFR and FFR data, from those patients with a phenotype of mostly microvascular dysfunction, to which there would be no benefit of coronary revascularization. Guidance for the assessment and diagnosis of microvascular angina has been provided by the COVADIS criteria [34]. Here, a diagnosis of microvascular angina can be made in the presence of 1. Symptoms of myocardial ischemia. 2. Absence of obstructive CAD (FFR > 0.8, lesion < 50% luminal reduction) by selective coronary angiography or CT coronary angiography. 3. Objective evidence of myocardial ischemia, and 4. Evidence of impaired coronary microvascular function [34]. For a further investigation of the coronary microcirculatory function, invasive testing utilizing thermodilution and intracoronary doppler may be performed to assess the coronary flow reserve, index of microvascular resistance, and pharmacologic challenge may be conducted for provocation of coronary vasospasm [35]. To further interrogate the coronary microcirculatory function, numerous modalities for non-invasive testing may be employed. These include coronary flow velocity ratio, as determined by transthoracic doppler echocardiography [36], positron emission tomography permitting the assessment of myocardial perfusion reserve and myocardial blood flow [37], cardiac MRI stress perfusion studies [38], and CT based perfusion assessment [39]. 

In the present review, data derived from case report data demonstrated a trend towards a phenotype demarcated by more diffuse coronary stenoses and coronary microvascular dysfunction in patients with CADASIL. As a result, the thorough investigation of the presence of obstructive CAD, objective measures of myocardial ischemia, and detailed assessment of the coronary microvascular function are prudent when signs and symptoms warrant further investigation, as treatment strategies may differ significantly.

## 7. Current Management of Stroke in CADASIL Patients

The current management of stroke and transient ischemic attacks (TIA) in CADASIL patients remains supportive, given that the most common manifestation is a lacunar stroke. In rare cases when acute strokes occur, intravenous thrombolysis has not been shown to be effective, for the cause is unlikely to be a thrombus that would be amenable to thrombolytic therapy [19].

Prevention of an event remains the mainstay management in CADASIL patients [1]. The strategic cornerstones of preventing stroke/TIA in CADASIL patients are similar to the non-CADASIL population, namely cholesterol modulation, blood pressure control, management of diabetes mellitus, weight control, and lifestyle modifications [1]. However, despite the lack of evidence, it is also not uncommon for CADASIL patients to be on antiplatelet therapy for the primary prevention of stroke and TIAs [1]. However, evidence is lacking regarding the use of antiplatelets, such as low-dose aspirin and clopidogrel, in the prevention of a primary stroke or coronary event [1]. Therefore, the administration of antiplatelets for the prevention of stroke or coronary events should be carefully weighed against the risk of intracerebral hemorrhage due to the high prevalence of cerebral microbleeds in this population [19].

## 8. The Risk of Intracranial Hemorrhage Associated with Antithrombotic and Antiplatelet Therapies

CADASIL is independently associated with an increased burden of cerebral micro-bleeds (CMB), which may increase the risk of intracranial hemorrhage (ICH) in the affected patients [40,41]. This may pose a dilemma for CADASIL patients requiring antithrombotic therapies for conditions such as acute coronary syndrome, atrial fibrillation, pulmonary embolism, or deep venous thrombosis.

Antiplatelet therapy, albeit not contraindicated, may increase the risk of ICH in patients with a high burden of CMB due to CADASIL [25,42]. However, it should be noted that there is no conclusive evidence to suggest the association between ICH and the use of dual antiplatelet therapy (DAPT). There are reports suggesting that CADASIL patients can be safely treated with DAPT for months without any significant increase in ICH; however, these were merely case reports without adequate follow-up [6,12].

Although not contraindicated, anticoagulants should be avoided in CADASIL patients, as they can increase the risk of ICH due to the high burden of CMB in these patients [9,28,43]. Therefore, this poses a challenging predicament for clinicians treating acute coronary syndromes in CADASIL patients. Whether screening for CMB burden with MRI would help stratify the risk of ICH in these patients remains a matter of further research.

## 9. Management of CAD in CADASIL Patients

### 9.1. Medical Therapy

There is no specific pharmacological therapy for the treatment of CAD in CADASIL patients. The main lines of treatment include symptom control, rehabilitation, and physiotherapy [3]. It is also advised to pursue strict control of modifiable cardiovascular risk factors such as arterial hypertension, diabetes mellitus, hypercholesterolemia, and cessation of smoking. There is no specific guidance regarding the use of antiplatelet therapy in these patients; therefore, in clinical practice, the same roles as for non-CADASIL patients are generally followed.

### 9.2. Optimal Myocardial Revascularization Strategy

Myocardial revascularization strategies are generally reserved for patients with flow-limiting obstructive coronary lesions. There is currently no clinical evidence to support the implementation of these invasive strategies in the context of ANOCA or coronary microvascular disease without the involvement of major epicardial coronary vessels. Owing to the unique nature of CAD in CADASIL patients and the absence of robust evidence, the optimal revascularization strategy should be discussed on an individual basis by a multidisciplinary heart team (Figure 1).

CADASIL patients with CAD requiring revascularization are typically younger individuals with a complex and aggressive pattern of disease [44]. In non-CADASIL patients, coronary artery bypass graft surgery (CABG) is favored in young patients with this type of complex disease, given its superior long-term outcomes as compared to percutaneous coronary intervention (PCI) [44]. However, in CADASIL patients, CABG may be associated with a higher risk of peri-operative ICH for two reasons. First, to facilitate the use of a cardiopulmonary bypass machine, large doses of heparin, typically around or in excess of 300 u/kg, are administered intraoperatively to achieve an activated clotting time greater than 480 s [45]. Second, CADASIL patients who have a higher risk of cerebrovascular accidents at baseline are more vulnerable to the effects of periodic hypotension and hemodynamic instability during the perioperative course of cardiac surgery [46]. Therefore, patients with CADASIL may have a higher risk of developing neurological events during the perioperative period. This is in accordance with the recommendations by the European Academy of Neurology in which hemodynamic swings, particularly hypotension, in the perioperative or periprocedural setting should be avoided to reduce the risk of cerebral ischemia [47]. Regarding the choice of the optimal coronary bypass conduit, there is no evidence to support or discourage the use of arterial grafts in these patients. It is unknown if CADASIL, a disease of small blood vessels, may affect the intimal layer of the internal mammary artery, the gold-standard conduit in CABG procedures.

In terms of PCI, multiple reports suggested its feasibility and safety in the context of CADASIL [10]. However, it is generally accepted that, in non-CADASIL patients, the long-term outcomes of PCI are inferior to CABG in young and diabetic patients with extensive CAD, particularly in terms of disease recurrence requiring the need for repeat revascularization [44]. This is attributed to the fact that coronary stents do not prevent or protect against the development of further coronary artery disease downstream of the coronary vessels [45]. This may constitute an additional challenge given the high tendency for the recurrence of CAD in CADASIL patients [44]. Additionally, in non-CADASIL patients, the presence of multiple stents necessitates the long-term use of DAPT for extended periods of time to prevent stent thrombosis; for CADASIL patients, long-term DAPT can theoretically further increase the risk of ICH in the long term, but it is of note that there are no studies exploring this association [4,23,25,43]. Nevertheless, PCI may offer an advantage over CABG in patients with CADASIL in terms of avoiding the periprocedural use of high doses of heparin and the hemodynamic instability that may occur in the peri-operative period.

### 9.3. The Role of Gene Therapy

In addition to conventional medical therapy and myocardial revascularization, the role of genetic counseling has been highlighted, especially for asymptomatic individuals who may be carrying the disease [3]. The main objective of current gene therapies is to prevent the accumulation of micro-aggregates in small blood vessels through various mechanisms, including the exclusion of *EGFR* from the *NOTCH3* protein, to prevent the development of ischemic lesions [48]. However, the adoption and validation of these therapies in the clinical setting remain a matter of further research.

## 10. Future Directions

Probably one of the most important areas to develop in the future is the creation of a reliable risk assessment and stratification model that can be used to predict the development and progress of CAD in high-risk CADASIL patients. Such a model would ideally compile data from different sources, such as the patient’s clinical data, genetic profile, and imaging modalities.

In addition, the pathophysiological mechanisms underpinning the development of extra-cranial vasculopathy, including CAD in CADASIL patients, remain unclear. Further insights into such vital mechanisms may result in new pathogenesis-directed therapies [49]. Currently, the accumulation of GOMs within the media of the arteries in the skin and the presence of *NOTCH3* are the diagnostic hallmarks of CADASIL syndrome [49]. However, the introduction of other markers of endothelial damage and repair can help with the understanding of endothelial dysfunction in these patients and its role in compromising dynamic blood flow responses. These potential biomarkers include plasma levels of von Willebrand factor and thrombomodulin. Also, blood levels of circulating progenitor cells have been associated with the severity and progress of the disease [49]. Strategies that prevent vascular dysfunction could also attenuate the risk of CAD in CADASIL patients. For instance, the use of *Notch3ECD* monoclonal antibodies, which bind to aggregates that compromise the vascular wall, has been shown to protect against cerebrovascular dysfunction. Also, the use of gene therapy to prevent the accumulation of protein aggregates in the vascular wall will probably play a major role in this group of patients [48]. Whether these potential therapies, albeit promising, would be beneficial in reducing the risk of developing CAD in these patients remains a matter of further research.

## 11. Study Limitations

There are some important limitations to this review. First, all the available studies are either case reports or small case series. In addition, the definition of significant CAD was not reported by most of the authors and relied mainly on visual estimation of the coronary lesions without further functional assessment. Finally, the details of optimal medical therapy or long-term follow-up of these patients were not reported by all authors.

## 12. Conclusions

Coronary artery disease in CADASIL patients is a serious condition that is characterized by its aggressive pattern and tendency to recur. In view of the lack of robust evidence, current therapeutic lines rely mainly on strict control of cardiovascular risk factors and symptom control. The optimal myocardial revascularization therapy, whether surgical or percutaneous, remains largely dependent on the patient’s clinical profile and the outcome of the multidisciplinary team discussion. Future efforts should be targeted toward the early detection and prevention of the disease progression, in addition to the introduction of novel targeted therapies.

## Figures and Tables

**Figure 1 medicina-59-00586-f001:**
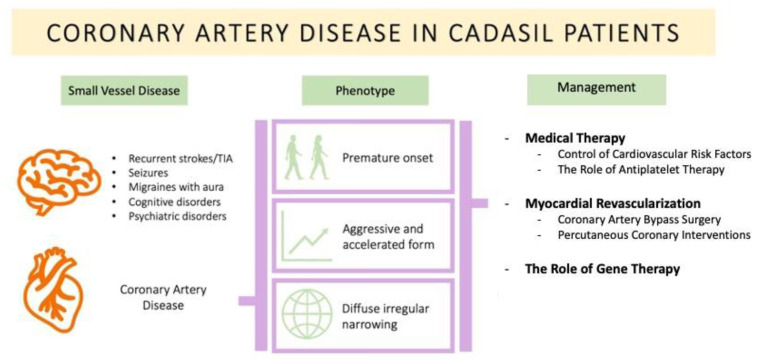
Overview of CAD in CADASIL patients. TIA: Transient Ischemic Attacks.

**Table 1 medicina-59-00586-t001:** Summary of case reports on CAD in CADASIL patients.

Name	Age at Presentation (Years)	Diagnosis	ECG Changes	Affected Coronary Territory
Rubin 2015 [6]	45	Unstable angina	Wellen’s T wave	1st episode: LAD 2nd episode: RCA
Briceno 2013 [8]	48	CAD	Sinus bradycardia	RCA
Tsanaxidis 2019 [9]	61	One vessel CAD	Anterior ST-Segment Elevation	LAD
Raghu 2003 [10]	30	Anterior MI	None specified changes	LAD
Langer 2020 [11]	54	ANOCA	Normal EKG	None with critical stenoses
Buczek 2016 [12]	57	Severe CAD	None specified changes	None specified

ANOCA: Angina with no obstructive coronary artery disease; ECG: Electrocardiogram; LAD: left anterior descending artery; MI: Myocardial Infarction; RCA: Right coronary artery.

## Data Availability

Not applicable.

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
