# Peer review of "Management of Coronary Artery Disease in CADASIL Patients: Review of Current Literature"

_medicina, 2023, doi:10.3390/medicina59030586_

Round 1

Reviewer 1 Report

We appreciate the opportunity to review your manuscript concerning CADASIL.

I have a few questions, and this manuscript's purpose is unclear even though it is a Review.

First, the paper states that the primary locus of disease in blood vessels is smooth muscle changes due to genetic factors. Rather than direct coronary artery narrowing, it is more likely that reduced blood flow in the micro-arteries affects myocardial ischemia. If this is true, there is no point in revascularization for coronary artery stenosis, and the issue should be devoted to genetic treatment for microarterial degeneration.

Also, the description of coronary artery stenosis is vague, and the treatment sites for PCI and CABG are not explicit. Especially if genetic degeneration of atherosclerosis is an issue, there should be a reference to the pros and cons of using arterial grafts in CABG. There is also no mention of whether stents may contribute to disease progression to coronary artery degeneration.

Rather than comparing the two treatments, I think a more realistic review would be to focus on the nature of the disease and state that coronary artery stenosis will be an essential issue in the future.

I think it is too abrupt to mention PCI and CABG in conclusion.

Author Response

Dear Reviewer,

Please find our point-by-point reply to your review. We will be very happy to answer any more questions if required.

General comments

  • We have increased the number of words to align with the journal’s policy
  • We have used track changes to facilitate the revision process
  • We have introduced new headings and sub-headings in the manuscript

Reply to Reviewer 1

- I have a few questions, and this manuscript's purpose is unclear even though it is a Review.

We thank the Reviewer; we have amended the manuscript (abstract and introduction sections) to clarify the purpose of this review

- First, the paper states that the primary locus of disease in blood vessels is smooth muscle changes due to genetic factors. Rather than direct coronary artery narrowing, it is more likely that reduced blood flow in the micro-arteries affects myocardial ischemia. If this is true, there is no point in revascularization for coronary artery stenosis, and the issue should be devoted to genetic treatment for microarterial degeneration.

We thank the Reviewer; we have amended the manuscript accordingly. We clarified that myocardial revascularization is indicated in presence of obstructive CAD. We also emphasised on the role of gene therapy to prevent the development and progress of this condition.

- Also, the description of coronary artery stenosis is vague, and the treatment sites for PCI and CABG are not explicit. Especially if genetic degeneration of atherosclerosis is an issue, there should be a reference to the pros and cons of using arterial grafts in CABG. There is also no mention of whether stents may contribute to disease progression to coronary artery degeneration.

We thank the Reviewer; we have amended the manuscript accordingly. We highlighted in the ‘study limitations’ section that the severity of CAD was reported based on visual estimation of the coronary lesions and that most studies lacked data from functional assessment of coronary lesions. We also discussed the use of arterial grafts the fats that coronary stenting does not prevent or protect against future coronary events.

- Rather than comparing the two treatments, I think a more realistic review would be to focus on the nature of the disease and state that coronary artery stenosis will be an essential issue in the future.

We thank the Reviewer; we have adopted this recommendation and adjusted the manuscript accordingly.

- I think it is too abrupt to mention PCI and CABG in conclusion.

We thank the Reviewer; we removed this from our conclusions section

Reviewer 2 Report

The manuscript by Servito et al. is interesting and well written. However, I have several suggestions:

 1.      In the Table 1, SCAD” is usually acronym for “spontaneous coronary artery dissection” rather than for “single vessel coronary artery disease”. This could be confused by readers.  Therefore, I suggest to replace “SCAD” with more appropriate acronym for “single vessel coronary artery disease”, for example “one vessel CAD”. Also, how did authors define “Severe CAD” in the Table 1?

2.      Could authors expand the subheading “Pathophysiology” with more detailed explanation regarding the cause of coronary microvascular dysfunction (CMD) in CADASIL-patients?

3.      The subheading “Cardiac manifestations of CADASIL” should be move before the subheading “Diagnosis”.

In this subheading, lines 129-130, the part of the sentence: “....evidence of distal fractional flow reserve less than 0.80, corresponding with the territories exhibiting subendocardial hypoperfusion” is not quite true, because it is well known that CMD leads to a higher value of FFR and therefore may underestimate the functional severity of the present epicardial coronary stenosis. I suggest to delete this part of sentence.

4.      The subheading “Diagnosis” should be expended with the role of invasive and noninvasive physiological tests in the detection of CMD and other CAD forms in CADASIL-patients.

5.      In the subheading “The risk of ICH with antithrombotic and antiplatelet therapies”, acronym “ICH should be replaced with the full name. 

In this subheading, acronym “CMB” was mentioned, but there is no explanation in the text what this acronym stands for.         

6.      The Figure 1 should be more colorful.

Author Response

Dear Reviewer,

Please find our point-by-point reply to your review. We will be very happy to answer any more questions if required.

General comments

  • We have increased the number of words to align with the journal’s policy
  • We have used track changes to facilitate the revision process
  • We have introduced new headings and sub-headings in the manuscript

Reply to Reviewer 2

The manuscript by Servito et al. is interesting and well written. However, I have several suggestions: 

  1. In the Table 1, „SCAD” is usually acronym for “spontaneous coronary artery dissection” rather than for “single vessel coronary artery disease”. This could be confused by readers.  Therefore, I suggest to replace “SCAD” with more appropriate acronym for “single vessel coronary artery disease”, for example “one vessel CAD”. Also, how did authors define “Severe CAD” in the Table 1?

We thank the Reviewer; we have amended the acronym as suggested. We have also mentioned in the “study limitations” section that most of the studies reported the severity of CAD based on visual estimation of the coronary lesions and that most studies lacked data from functional assessment of coronary lesions.

  1. Could authors expand the subheading “Pathophysiology” with more detailed explanation regarding the cause of coronary microvascular dysfunction (CMD) in CADASIL-patients?

We thank the Reviewer; we have amended the manuscript accordingly.

  1. The subheading “Cardiac manifestations of CADASIL” should be move before the subheading “Diagnosis”.

We thank the Reviewer; we have amended the manuscript accordingly

In this subheading, lines 129-130, the part of the sentence: “....evidence of distal fractional flow reserve less than 0.80, corresponding with the territories exhibiting subendocardial hypoperfusion” is not quite true, because it is well known that CMD leads to a higher value of FFR and therefore may underestimate the functional severity of the present epicardial coronary stenosis. I suggest to delete this part of sentence.

We thank the Reviewer; we have deleted this part of the sentence.

  1. The subheading “Diagnosis” should be expended with the role of invasive and non-invasive physiological tests in the detection of CMD and other CAD forms in CADASIL-patients.

We thank the Reviewer; we have amended the manuscript accordingly. We have added an extensive section on the diagnosis of CAD especially in the context of CMD.

  1. In the subheading “The risk of ICH with antithrombotic and antiplatelet therapies”, acronym “ICH should be replaced with the full name.  In this subheading, acronym “CMB” was mentioned, but there is no explanation in the text what this acronym stands for.          

We thank the Reviewer; we have amended the manuscript accordingly.

  1. The Figure 1 should be more colorful.

We thank the Reviewer; we have introduced a new figure

Round 2

Reviewer 1 Report

Your efforts have made the Review more accessible and easier to read.

We appreciate your work.

Since the conclusion of your manuscript has changed, I think you should omit the diagram of the balance between PCI and CABG in Figure 1.

Do you have any typical coronary angiograms and pathology photos that you can provide? I think that would be more useful to the reader.

Author Response

Dear Reviewer,

Thank you very much for your edits/suggestions.

Your efforts have made the Review more accessible and easier to read.

We appreciate your work.

Since the conclusion of your manuscript has changed, I think you should omit the diagram

of the balance between PCI and CABG in Figure 1.

We have amended the figure. We removed the comparison between PCI and CABG and included general lines of management instead.

Do you have any typical coronary angiograms and pathology photos that you can provide? I think that would be more useful to the reader.

Unfortunately we have no angiograms or photos that we can use.

Reviewer 2 Report

The manuscript could be accepted in the present form.

There are several typos in the manuscript:

1.      Line 49, instead of „CADA“, it should stand „CAD“.

2.      Line 53, instead of „patten“, it should stand „pattern“.

3.      In the Table 1, abbreviations „CAD“ and „RCA“ should be explained in the figure legend. Also, instead of „EKG“, it should stand „ECG“.

4.      Line 235, instead of „coronary artery disease“, it should stand „CAD“.

Author Response

Dear Reviewer,

Thank you very much for your edits/suggestions. We have made all the suggested edits/suggestions and we ran a full linguistic / grammar check.

There are several typos in the manuscript:

  1. Line 49, instead of „CADA“, it should stand „CAD“.
  2. Line 53, instead of „patten“, it should stand „pattern“.
  3. In the Table 1, abbreviations „CAD“ and „RCA“ should be explained in the figure legend. Also, instead of „EKG“, it should stand „ECG“.
  4. Line 235, instead of „coronary artery disease“, it should stand „CAD“.